# Spatial transcriptomics of meningeal inflammation reveals inflammatory gene signatures in adjacent brain parenchyma

Sachin P Gadani[1,2†], Saumitra Singh[1†], Sophia Kim[1], Jingwen Hu[1], Matthew D Smith[1], Peter A Calabresi[1,3], Pavan Bhargava[1]*

[1]Division of Neuroimmunology, Department of Neurology, Johns Hopkins University School of Medicine, Baltimore, United States; [2]Department of Neurology, University of Pittsburgh, Pittsburgh, United States; [3]Solomon Snyder, Department of Neuroscience, Johns Hopkins University School of Medicine, Baltimore, United States

*For correspondence: pbharga2@jhmi.edu

†These authors contributed equally to this work

Competing interest: The authors declare that no competing interests exist.

## eLife Assessment

Brain inflammation is a hallmark of multiple sclerosis. Using novel spatial transcriptomics methods, the authors provide **solid** evidence for a gradient of immune genes and inflammatory markers from the meninges toward the adjacent brain parenchyma in a mouse model. This **important** study advances our understanding of the mechanisms of brain damage in this autoimmune disease.

**Abstract** While modern high efficacy disease modifying therapies have revolutionized the treatment of relapsing-remitting multiple sclerosis, they are less effective at controlling progressive forms of the disease. Meningeal inflammation is a recognized risk factor for cortical gray matter pathology which can result in disabling symptoms such as cognitive impairment and depression, but the mechanisms linking meningeal inflammation and gray matter pathology remain unclear. Here, we performed magnetic resonance imaging (MRI)-guided spatial transcriptomics in a mouse model of autoimmune meningeal inflammation to characterize the transcriptional signature in areas of meningeal inflammation and the underlying brain parenchyma. We found broadly increased activity of inflammatory signaling pathways at sites of meningeal inflammation, but only a subset of these pathways active in the adjacent brain parenchyma. Subclustering of regions adjacent to meningeal inflammation revealed the subset of immune programs induced in brain parenchyma, notably complement signaling and antigen processing/presentation. Trajectory gene and gene set modeling analysis confirmed variable penetration of immune signatures originating from meningeal inflammation into the adjacent brain tissue. This work contributes a valuable data resource to the field, provides the first detailed spatial transcriptomic characterization in a model of meningeal inflammation, and highlights several candidate pathways in the pathogenesis of gray matter pathology.

## Introduction

Multiple sclerosis (MS) is a chronic autoimmune disease of the central nervous system (CNS) characterized by a relapsing remitting and/or progressive course of demyelination, axonal injury, and neurologic dysfunction (*Reich et al., 2018*). Highly efficacious disease modifying therapies have revolutionized the prevention and treatment of MS relapses, but are less effective at stopping hallmarks of MS progression such as brain atrophy (*Faissner et al., 2019*). Accumulating evidence points to a pivotal role for leptomeningeal inflammation (LMI) in contributing to this pathology (*Howell et al., 2011*; *Absinta et al., 2015*). LMI is found in all subtypes of MS, ranging histologically from disorganized

collections of leukocytes to highly organized ectopic lymphoid follicles, and correlates with the presence of cortical gray matter demyelination, neurite loss, and decreased volume (*Wicken et al., 2018*). Gray matter pathology (GMP) has been linked to debilitating symptoms such as cognitive impairment and depression (*Geurts and Barkhof, 2008*), and tends to occur in spatial relation to areas of LMI. Indeed, the most common gray matter lesion location is directly sub-pial (*Bø et al., 2003a*), and there is a gradient of increased pathology toward the surface of the brain in MS patients (*Magliozzi et al., 2010*; *Mainero et al., 2015*). Interestingly, GMP occurs without local blood-brain barrier disruption or robust infiltration of peripheral immune cells into the brain parenchyma (*van Horssen et al., 2007*; *Bø et al., 2003b*). LMI is therefore speculated to be a source of pro-inflammatory molecules that contribute to GMP (*Pikor et al., 2015*), but the pathway(s) involved remain(s) unknown.

Several putative mechanisms linking LMI to GMP have been proposed. Magliozzi and colleagues identified increased expression of numerous cytokines and chemokines, including interferon gamma (IFNγ), tumor necrosis factor (TNF), interleukin (IL)-2, IL-22, CXCL13, and CXCL10, in the meninges and CSF of post-mortem MS cases with high levels of meningeal inflammation and GM demyelination (*Magliozzi et al., 2018*). Microarray analysis of cortical lesions from MS cases with LMI revealed a shift in TNF signaling from TNFR1/TNFR2 and I-mediated anti-apoptotic pathways toward TNFR1- and RIPK3-mediated pro-apoptotic/pro-necroptotic pathways (*Magliozzi et al., 2019*). In marmoset and rat models of experimental autoimmune encephalomyelitis (EAE), sub-pial cortical lesions were found with prominent microglial activation and immunoglobulin deposition on myelin sheaths (*Storch et al., 2006*; *Merkler et al., 2006*). Interestingly, complement deposition, one possible mechanism of immunoglobulin-related cellular injury, is not found in purely cortical gray matter lesions, in contrast to white or gray/white matter lesions (*Brink et al., 2005*). Numerous other mediators of injury, including reactive oxygen and nitrogen species, metabolic stress, and excitotoxicity have been proposed to drive neurodegeneration in MS and may be at play in GMP.

Previous attempts to characterize the relation between LMI and GMP have been limited by the absence of spatially resolved data, which therefore lacks critical information about the anatomic relationship between LMI and the underlying brain parenchyma. Here, we present a spatial transcriptomic dataset and analysis in a mouse model of relapsing/remitting CNS autoimmunity and meningeal inflammation, Swiss Jim Lambert (SJL) mouse EAE (*Magliozzi et al., 2004*). Our prior work in SJL EAE demonstrated meningeal areas of gadolinium contrast enhancement on magnetic resonance imaging (MRI) that correspond histologically to collections of B cells, T cells, and myeloid cells (*Bhargava et al., 2021*). In the parenchyma adjacent to areas of meningeal inflammation we identified astrocytosis, activated microglia, demyelination, and evidence of axonal stress and damage (*Bhargava et al., 2021*). This work characterizes the spatially resolved transcriptome of LMI in the SJL EAE model system, revealing the subset of genes and gene sets active in the LMI that extend into the adjacent parenchyma and providing insights into immune pathways that could underlie sub-pial neurodegeneration.

## Results

We tracked the development of LMI during SJL EAE using contrast-enhanced serial MRI imaging. MRI data was then used to target areas of LMI for spatial transcriptomics (*Figure 1A*). Mice developed a characteristic relapsing pattern of neurologic impairment, and contrast-enhanced MRI was performed at weeks 6, 8, and 10 post-immunization (*Figure 1B and C*). In the SJL EAE model, contrast enhancing meningeal lesions are most frequently found in the intrapeduncular cistern, quadrigeminal cistern, and the cleft between the hippocampus and medial geniculate nucleus (*Bhargava et al., 2021*; *Bedussi et al., 2017*; *Figure 1D*). Lesion number remained stable throughout the disease course independent of EAE score (*Figure 1—figure supplement 1A and B*).

Brain slices were collected from four naïve mice and four EAE mice 11 weeks post-immunization and used to prepare five and six samples for spatial transcriptomics, respectively. Hematoxylin and eosin (H&E) staining confirmed the presence of meningeal inflammation in the areas of contrast enhancement, as we had previously observed (*Bhargava et al., 2021*; *Figure 1E*). Since naïve animals were used as controls, we confirmed that CFA alone does not produce lasting glial reactivity or LMI formation. Groups of animals were given CFA only or left naïve. Neither group developed neurologic signs, and after 11 weeks the brains were processed for IHC analysis. There was no evidence of LMI development, and no difference in glial reactivity as measured by GFAP, IBA1, or CD68 intensity (*Figure 1—figure supplement 1C–E*).

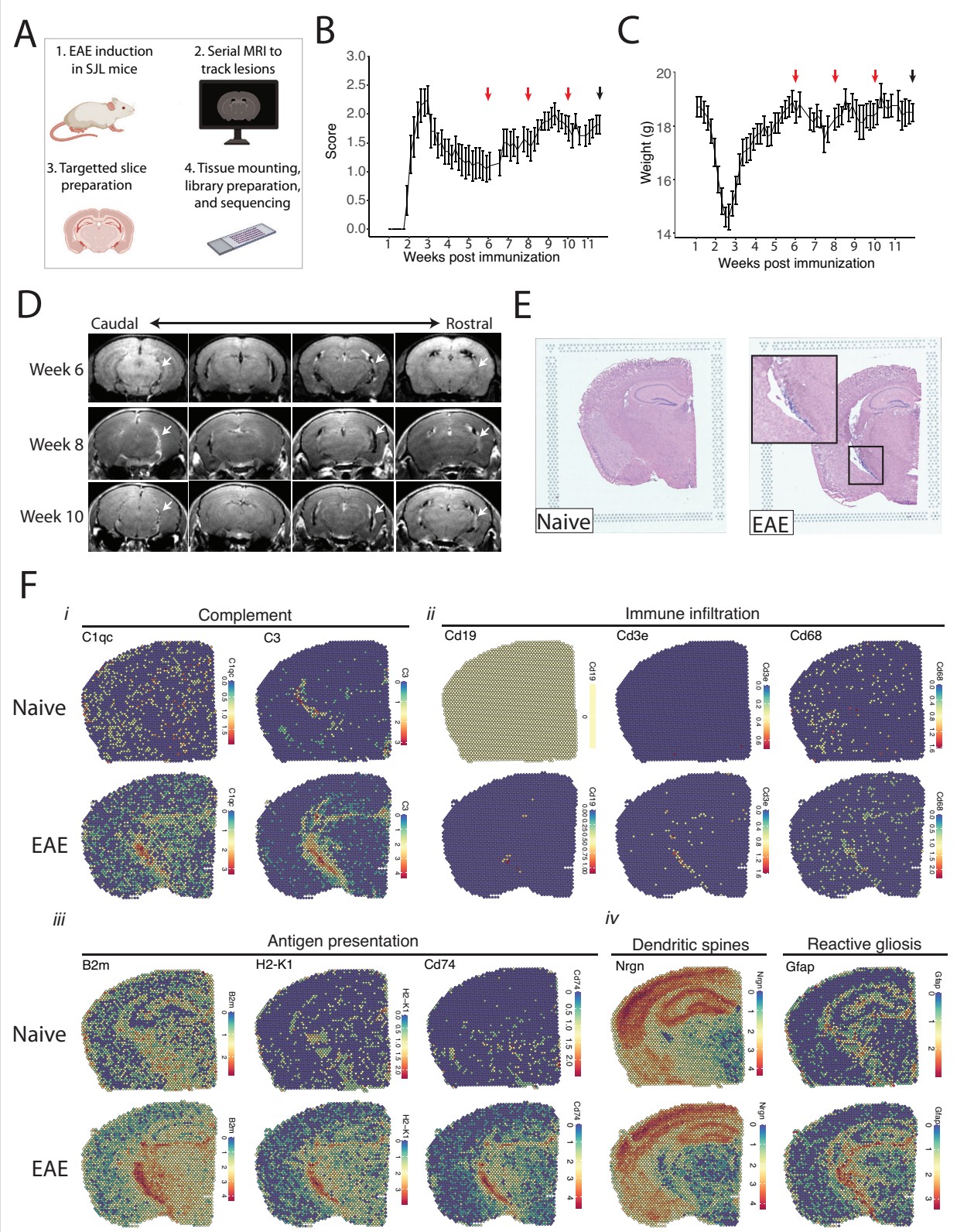

**Figure 1.** Magnetic resonance imaging (MRI)-guided spatial transcriptomics of meningeal-based inflammation in Swiss Jim Lambert (SJL) experimental autoimmune encephalomyelitis (EAE). (**A**) Schematic describing the experimental paradigm. SJL mice underwent brain MRI 6, 8, and 10 weeks' post-immunization with PLP$_{139-151}$. Brain slices from regions with meningeal inflammation were collected and processed for spatial transcriptomics on the 10x Genomics platform. (**B–C**) Behavior scores (**B**) and mouse weights (**C**) of the EAE cohort. Red arrows indicate MRI time points, black arrow indicates

*Figure 1 continued on next page*

*Figure 1 continued*

time of tissue harvesting (N=4). (**D**) Representative post-contrast MRI brain images, white arrows indicate areas of meningeal-based inflammation. (**E**) Representative images of hematoxylin and eosin (H&E)-stained tissue sections mounted on spatial transcriptomics slides (left, naïve; right, EAE). (**F**) Spatial feature plots from naïve (top row) and EAE (bottom row) representative samples demonstrate altered expression of genes related to complement (**i**), immune infiltration (**ii**), antigen presentation (**iii**), dendritic spines, and astrocyte activation (**iv**).

The online version of this article includes the following figure supplement(s) for figure 1:

**Figure supplement 1.** Contrast enhancing meningeal inflammation in Swiss Jim Lambert (SJL) experimental autoimmune encephalomyelitis (EAE) does not change at chronic time points or with clinical disease scores.

**Figure supplement 2.** Quality control analysis of spatial transcriptomic data.

Spatial transcriptomic data from all samples were of high quality and read depth when assessed by treatment group (*Figure 1—figure supplement 2A–C*) or sample (*Figure 1—figure supplement 2D and E*). There was expected anatomic variability in number of read counts and number of features per spot, with relatively low counts and features in white matter as compared to the cortex or hippocampus (*Figure 1—figure supplement 2F and G*), and spots had similar degrees of average complexity between EAE and naïve samples (*Figure 1—figure supplement 2H*). UMAP clustering revealed no significant independent effect of sample (*Figure 1—figure supplement 2I*) or slide (*Figure 1—figure supplement 2J*). Numerous genes were differentially expressed between EAE and naïve slices as estimated by DESeq2 on samples pseudobulked by biological replicate (*Supplementary file 1*; *Figure 1—figure supplement 2K*), including genes associated with the complement cascade, immune infiltration, antigen presentation, and astrocyte activation (*Figure 1F*; *Figure 1—figure supplement 2L*).

We next explored the activity of a broad range of pathways using the pathway responsive genes (PROGENy) method (*Schubert et al., 2018*; *Figure 2A*). Inflammatory pathways related to TNF, JAK-STAT, and NFκB signaling were upregulated in EAE compared to naïve, with peak activity around sites of meningeal inflammation (*Figure 2B and C*). Pathways related to Trail and PI3K were downregulated in EAE compared to naïve (*Figure 2D and E*), and TGFβ pathway activity was unchanged between groups (*Figure 2D and E*).

To focus our analyses on foci of meningeal inflammation specifically, we performed unbiased UMAP clustering on the spatial transcriptomic dataset and identified 12 distinct clusters (*Figure 3A*). Grouping the dimensional reduction UMAP plot by EAE and naïve revealed that cluster eleven (C11) was restricted to EAE samples (*Figure 3B*). C11 makes up 1–5% of the total spots in those samples (*Figure 3C*), or about 20–120 total spots, and was significantly enriched in EAE relative to naïve samples (*Figure 3D*). Visualizing the clusters using spatial feature plots confirmed that most clusters map consistently to specific anatomic regions and were similar between naïve and EAE (*Figure 3E*; *Figure 3—figure supplement 1A*). C11 overlapped with previously noted areas of meningeal MRI enhancement, suggesting that C11 represents areas of meningeal inflammation (*Figure 3E*; *Figure 3—figure supplement 1B—D*). We compared the gene expression in C11 to other clusters and found 132 upregulated genes and 70 downregulated genes (p<0.05; log 2 fold change >1; *Figure 3F*). Inflammation-related genes were prominently represented in C11 relative to other clusters (*Supplementary file 2*), with top genes including *Cd74*, *C3*, and *Gfap* (*Figure 3G*; *Figure 3—figure supplement 1E*). The presence of glial genes such as *Gfap* within cluster 11 likely represents areas of sub-pial brain parenchyma included within the cluster. Immunoglobulin genes, which are highly expressed in areas of human GMP (*Magliozzi et al., 2019*), were also upregulated in C11 (*Figure 3—figure supplement 1F*). We next performed gene set enrichment analysis of C11 spots using the gene ontology (GO) database (*Carbon et al., 2021*; *Ashburner et al., 2000*), finding 538 variable gene sets (adjusted p-value<0.05; *Supplementary file 3*). Among the most prominently enriched gene sets were those involved in antigen processing and presentation, complement activation, lymphocyte activation, and cytokine production and response (*Figure 3H*).

After establishing the defining transcriptomic features of meningeal inflammation in our model, we next sought to characterize inflammatory changes in the adjacent CNS parenchyma. We performed unbiased subclustering of each cluster in turn and found EAE-specific subclusters and differentially expressed genes within clusters 1 and 2. These subclusters were labeled 1_3, 1_4, and 2_6 (*Figure 4A and B*; *Figure 4—figure supplement 1*) and represent regions of the thalamus and hypothalamus (*Figure 4C*).

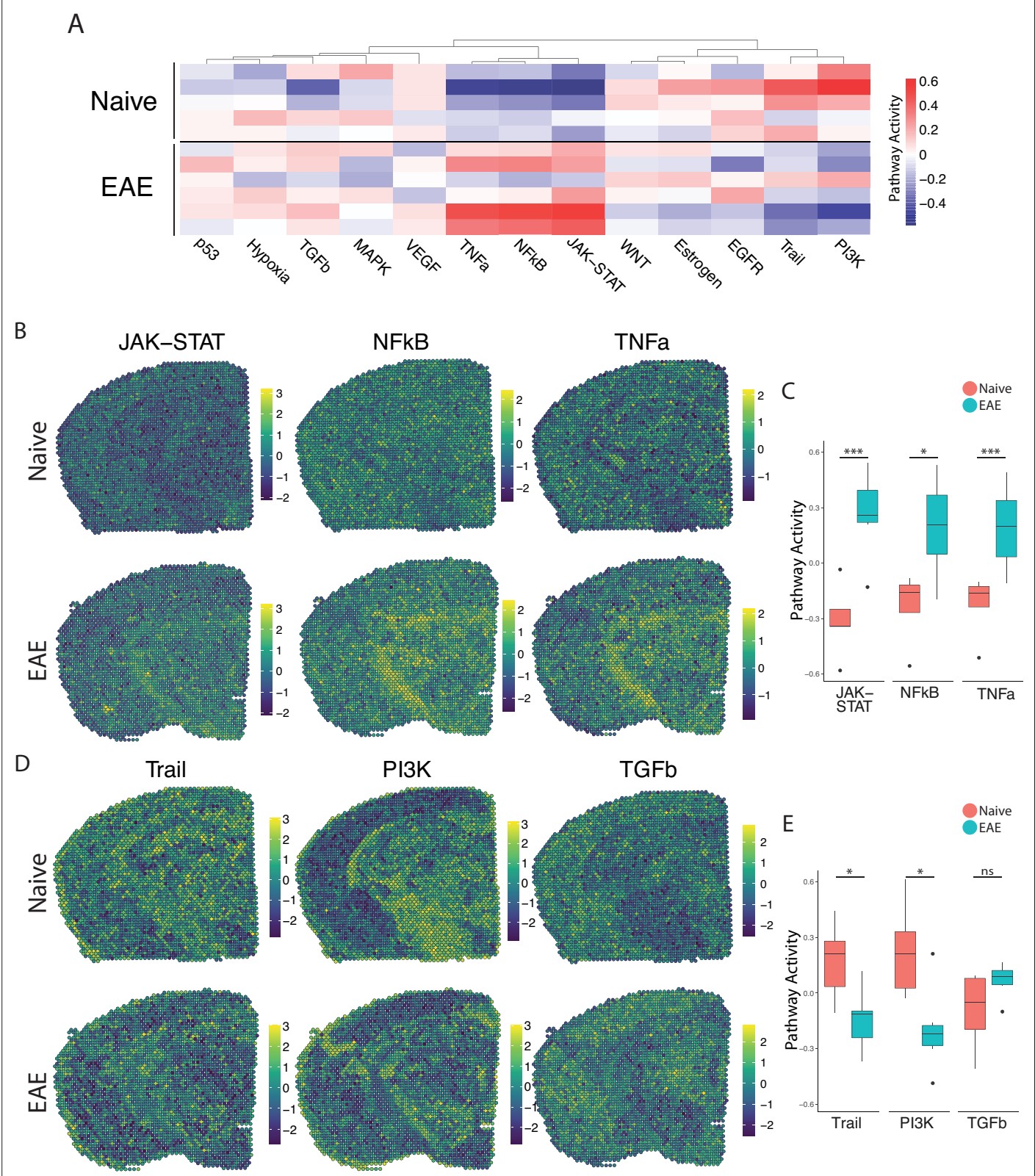

**Figure 2.** PROGENy analysis reveals spatially restricted pathway activity differences between naïve and experimental autoimmune encephalomyelitis (EAE). (**A**) Heatmap displaying averaged PROGENy pathway analysis results. (**B**) Representative spatial plot showing activity of the JAK-STAT, NFkB, and TNFa signaling pathways. (**C**) Comparison of JAK-STAT, NFkB, and TNFa pathway activities between groups. (**D**) Representative spatial plot showing activity of the Trail, PI3K, and TGFb signaling pathways. (**E**) Comparison of Trail, PI3K, and TGFb pathway activities between groups. (Naïve mouse N=4,

*Figure 2 continued on next page*

*Figure 2 continued*

sample N=5; EAE mouse N=4, sample N=6; multiple t-tests corrected for multiple comparisons with the Benjamini, Krieger, and Yekutieli method; *p<0.05, **p<0.01, ***p<0.001.)

Next, we assessed whether EAE-specific subclusters were physically closer to meningeal inflammation than other related subclusters. The distance between the average location in each subcluster to the nearest point of C11 was calculated. In the subclusters of cluster 1, we found no difference in proximity to C11 between EAE-specific subclusters vs. subclusters present in EAE and naïve. However, in subclusters of cluster 2, the EAE-specific subcluster was significantly closer to C11 on average compared to other subclusters (*Figure 4D*). To explore which pathways were activated in these subclusters, we performed gene set enrichment analysis using the GO database (*Figure 4E–G*; *Supplementary file 4*, *Supplementary file 5*, *Supplementary file 6*). All subclusters displayed enrichment of GO gene sets related to inflammation, antigen processing and presentation, and humoral immunity. Interestingly, pathways related to neuron death, cellular stress, and negative regulation of cellular processes were also enriched (*Figure 4E–G*; *Figure 4—figure supplement 2*). Interestingly, other pathways related to cell death, including positive regulation of apoptotic process or programmed cell death, were upregulated in subcluster 1_3 but downregulated in subcluster 1_4, suggesting spatially variable survival signals (*Supplementary file 4*, *Supplementary file 5*, *Supplementary file 6*). Overall, 31 upregulated pathways were conserved between cluster 11 and subclusters 1_3, 1_4, 2_6 (*Figure 4H*). We found prominent conserved upregulation of pathways related to adaptive/B-cell-mediated immunity, antigen processing and presentation, cell killing, and others (*Figure 4I*).

Our pathway analysis of meningeal inflammation and areas of inflamed adjacent brain parenchyma suggested that inflammatory signals increased in meningeal inflammation could have variable 'penetration' into the adjacent brain. We sought to test this using spatial trajectory gene/gene set expression modeling available within the SPATA2 software package (*Kueckelhaus et al., 2020*). Trajectories were drawn in EAE samples from the largest region of meningeal inflammation to the central thalamus (*Figure 5A*). Gene and gene set expression levels were then evaluated along the length of these trajectories and compared to ideal patterns of expression, as demonstrated for representative genes *B2m* and *C3* (*Figure 5B–E*). The difference between gene or gene set expression and the ideal patterns, here 'logarithmic descending' or 'gradient descending', is represented by the residual line (*Figure 5B and E*). The area under the residual curve (residual AUC) is therefore inversely proportional to fit for the given gene or gene set and ideal pattern. *C3* expression declines rapidly along the trajectory and the logarithmic descending residual AUC is lower than gradient descending residual AUC (*Figure 5E*), while *B2m* follows a less steep decline and fits the two patterns similarly (*Figure 5C*). Trajectory analysis of gene sets enriched in meningeal inflammation and adjacent brain parenchyma was performed in this way, and the average residual AUC was calculated for gradient descending/ascending and logarithmic descending/ascending patterns (*Figure 5Fi*). As expected, all gene sets fit descending patterns better than ascending ones. There was variability in fit to the gradient descending pattern of expression, representing a more gradual decline in pathway enrichment along the trajectory, with gene sets related to antigen processing and presentation, cell killing, IL-6 production, and IFNγ response having the best fit (*Figure 5Fi*). Enrichment score trajectory heatmap of a representative sample corroborates this, showing increased activity farther along the spatial trajectory for these gene sets (*Figure 5Fii*).

We next validated this variability in pattern of expression in a separate cohort of SJL mice with EAE using RNAscope to label selected transcripts related to glial activation (*Fcgr3* and *Gfap*) antigen presentation (*B2m* and *Cd74*) and complement (*C3*) (*Figure 5G and I*). Consistent with our spatial transcriptomics data, each of these transcripts was substantially induced in EAE as compared to naïve (*Figure 5—figure supplement 1*). We quantified the shortest distance of each target-positive cell from the region of LMI (*Figure 5H and J*). For all transcripts, the number of positive cells decreased with distance exponentially. Exponential regression was applied to find the best fit curves of the data, modeled by the equation $y=a * x^b$, where y is percentage of cells, x is distance from LMI. The exponential constant b reflects the rate of decrement and had higher absolute value in *C3* (b=−1.13) relative to *B2m* (b=−0.59) (*Supplementary file 7*).

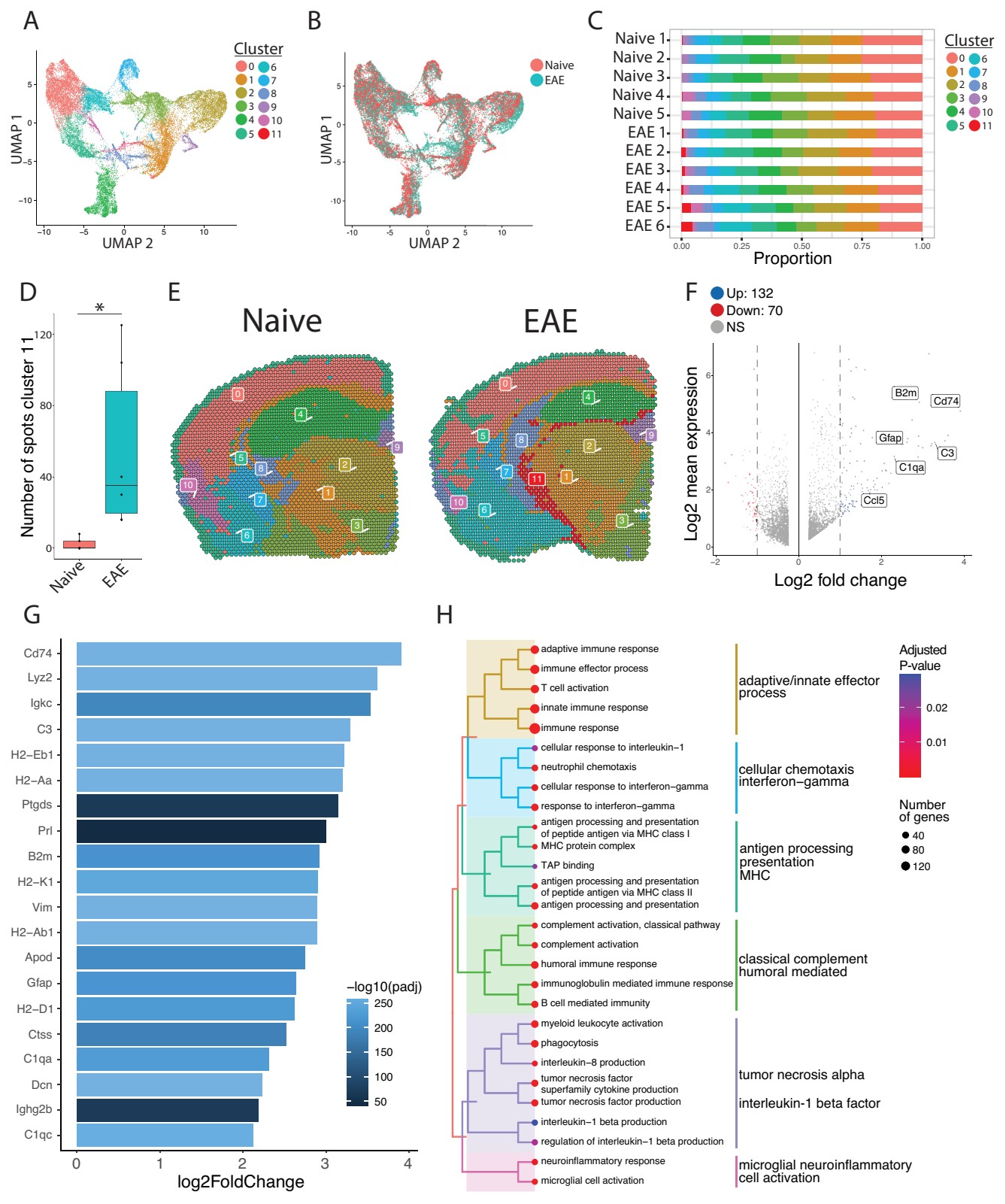

**Figure 3.** Unbiased clustering reveals a group of spots enriched in inflammatory genes. (A–B) UMAP dimensionality reduction plots colored by (**A**) cluster or (**B**) group. (**C**) Bar plot showing the proportion of spots in each cluster by sample. (**D**) Number of spots in cluster 11 by group (N=11; Student's two-tailed t-test). (**E**) Representative spatial feature plots of naïve and experimental autoimmune encephalomyelitis (EAE) samples showing the spatial distribution of each cluster. (**F**) MA plot comparing differences in gene expression between cluster 11 and all other clusters averaged across

*Figure 3 continued on next page*

*Figure 3 continued*

samples. Red and blue spots represent genes in cluster 11 that are significantly increased or decreased, respectively (adjusted p-value<0.05, log 2 fold change >1). (**G**) Bar plot of top 15 genes enriched in cluster 11 compared to other clusters. (**H**) Tree plot displaying gene set enrichment results using the gene ontology (GO) database. Spots in cluster 11 were compared to other spots and gene set sizes ranging from 10 to 500 were included (adjusted p-value<0.05).

The online version of this article includes the following figure supplement(s) for figure 3:

**Figure supplement 1.** Spatial and transcriptional properties of inflammatory clusters.

## Discussion

Meninges-restricted inflammation in MS is intimately related to sub-pial gray matter demyelination (*Howell et al., 2011*), atrophy, and neurocognitive symptoms, but therapeutically targeting this aspect of the disease is challenging due to poorly understood pathologic mechanisms. Here, we present spatial transcriptomics analysis in a mouse model of LMI, finding a broad swath of inflammatory pathways upregulated at foci of LMI and a subset of them upregulated in the nearby brain parenchyma. Notably, genes related to B-cell-mediated responses and antigen processing and presentation were upregulated in inflammatory parenchymal subclusters. Variable penetrance of inflammatory pathway activity was evident when analyzing the expression pattern of gene sets along a linear trajectory from LMI into the CNS parenchyma, where antigen processing and presentation, IL-6 production, and the IFNγ response pathway activity followed a more gradual decline as compared to other pathways.

An important limitation of the spatial transcriptomics method used in this study is the spatial resolution of each spot. Since each spot is ~55 μm in diameter it is likely that multiple cells are captured in each one. The simultaneous addition of single-cell analysis would facilitate deconvolution of cell types within each spot. Furthermore, some spots of the borders of anatomic structures are likely to include both regions. This is exemplified by the spots within cluster 11, which in this study mainly represent meningeal inflammation, also being enriched in glial genes such as *Gfap*. Other methods of spatial RNA analysis exist that allow single-cell resolution, but to date are probe-based and therefore limited by the number of genes assessed.

Prior work supports roles for pathways identified in our dataset, including B cell and IFNγ-mediated responses, in contributing to neurodegeneration in MS. LMI in MS is rich in B cells, and the critical role of B cells in MS has been underscored by the success of B cell depletion in relapsing and progressive MS. Recent studies have proposed numerous mechanisms whereby B cells could contribute to cortical pathology, including indirectly through activation and inflammatory polarization of T cells, myeloid cells, and astrocytes or directly through production of neurotoxic cytokines or antibodies (*Bhargava et al., 2022*). B cell culture supernatants from MS patients, but not healthy controls, are toxic to rat and human neurons and oligodendrocytes with this effect being mediated by the extracellular vesicle fraction of the supernatants (*Lisak et al., 2017*; *Benjamins et al., 2019*). B cells are also sources for inflammatory cytokines, such as IL-6 and GM-CSF, and antibodies, which are speculated to contribute to GMP (*Bhargava et al., 2022*).

The role of IFNγ in the pathogenesis of MS and EAE is complex, and likely has stage-specific protective and pathologic effects (*Arellano et al., 2015*). Its prominent upregulation in EAE was initially considered evidence of its pathogenic nature, but subsequent experiments showed that it was pathogenic during the initiation phase but protective later in the course (*Arellano et al., 2015*). IFNγ signaling and subsequent upregulation of antigen processing and presentation on glia has been identified as a mechanism that could lead to remyelination failure and subsequent neuronal loss (*Kirby et al., 2019*). Oligodendrocyte precursor cells upregulate antigen presentation and cross-presentation pathways in response to IFNγ, promoting inflammation and making them susceptible to CD8+ T cell killing (*Kirby et al., 2019*). Upregulation of genes for self-antigen presentation has also been noted in neurons and oligodendrocyte lineage (OL) cells in post-mortem MS single nucleus RNAseq (*Schirmer et al., 2019*) and OL cells in EAE (*Falcão et al., 2018*; *Jäkel et al., 2019*; *Langseth et al., 2023*; *Kukanja et al., 2024*). Recent work also assessed subcortical white matter lesions from post-mortem MS cases via spatial transcriptomics, finding prominent elevation of TNF signaling, in agreement with presented results in the SJL EAE model (*Lerma-Martin et al., 2022*).

Another important consideration in these experiments is our choice of naïve, rather than CFA only, controls. While often used as the control in EAE studies focused on mechanisms of autoimmunity,

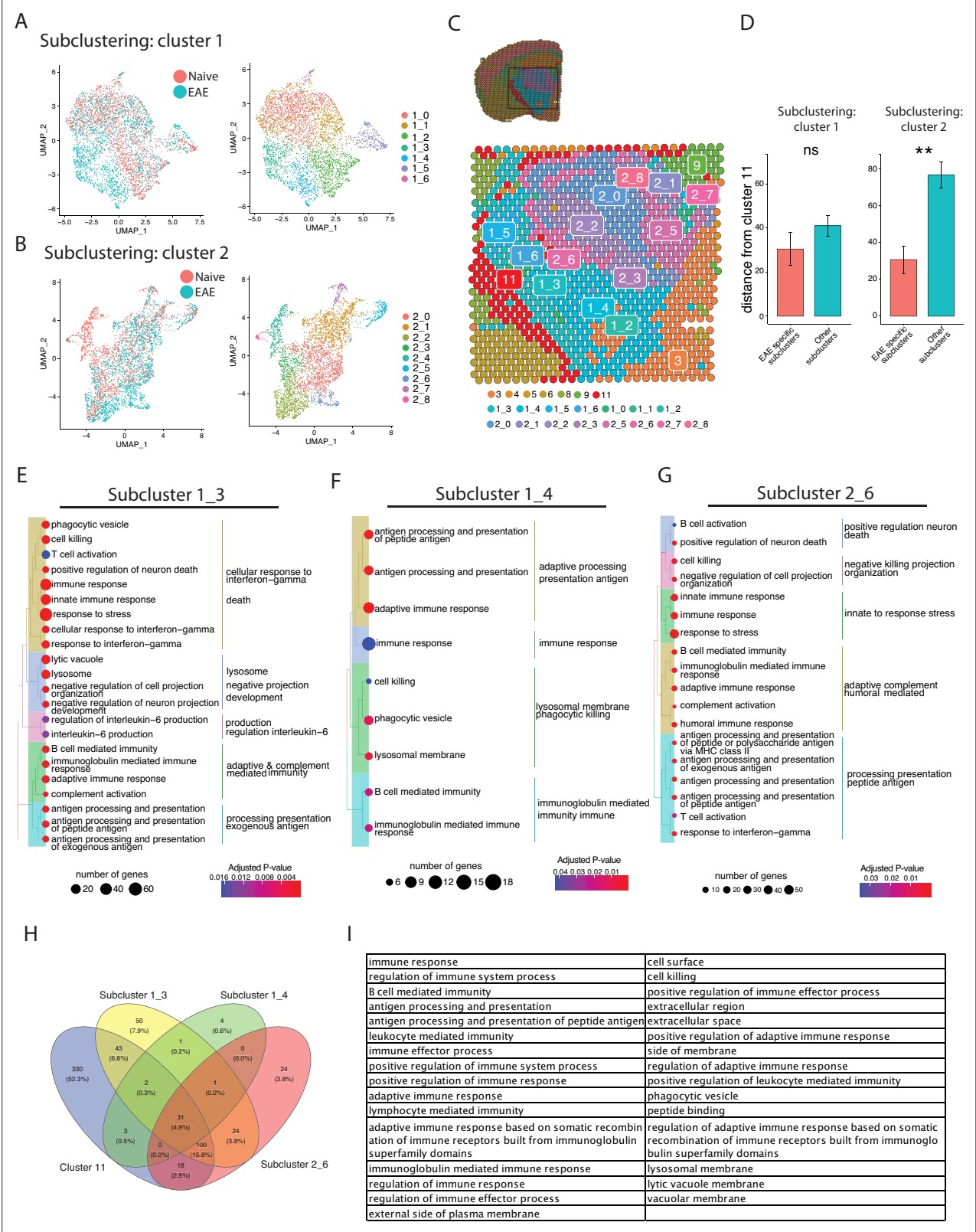

**Figure 4.** Subclustering of spots adjacent to meningeal immune follicles reveals a subset of active immune patterns. (**A**) UMAP dimensionality reduction plots showing subclustering of cluster 1 colored by (left) group or (right) cluster. (**B**) UMAP dimensionality reduction plots showing subclustering of cluster 2 colored by (left) group or (right) cluster. (**C**) Representative spatial feature plot showing the locations of cluster 1 and 2 subclusters. (**D**) Distance from the center of indicated subclusters to the nearest point of cluster 11 (N=11; Student's two-tailed t-test). (E–G) Tree plot displaying gene set

*Figure 4 continued*

enrichment results using the gene ontology (GO) database for subcluster 1_3 (**E**), 1_4 (**F**), and 2_6 (**G**) compared to other spots in their respective clusters. (**H**) Venn diagram shows overlap of significantly enriched GO gene sets between cluster 11 and subclusters 1_3, 1_4, and 2_6, with (**I**) 31 gene sets elevated in all. GO gene set of size ranging from 10 to 500 were included (adjusted p-value<0.05).

The online version of this article includes the following figure supplement(s) for figure 4:

**Figure supplement 1.** Subcluster analysis reveals experimental autoimmune encephalomyelitis (EAE)-specific subclusters and gene enrichment.

**Figure supplement 2.** Gene contributions to enriched gene sets in subclusters 1_3, 1_4, and 2_6.

CFA only can independently induce systemic inflammation. Since this study seeks to describe transcriptomic changes in neuroinflammation more broadly, we chose to use a healthy comparison group to maximize our ability to find genes enriched in neuroinflammation. Ultimately, however, the choice of naïve or CFA only controls is unlikely to have affected our conclusions. SJL-EAE, unlike the more common C57Bl6-EAE, does not require pertussis toxin during the induction. The only difference between naïve and CFA only controls is the subcutaneous CFA delivered at time of immunization (11 weeks prior to experiment endpoint). Indeed, when we compared CFA only and healthy animals at 11 weeks there was no difference in glial reactivity by GFAP, IBA1, or CD68 mean fluorescence intensity (MFI). There was also no evidence of neurologic symptoms or LMI development in CFA only controls.

While SJL EAE models many features of LMI in MS (*Magliozzi et al., 2004*), there are important differences that limit the direct translation of these results to MS. The majority of LMI identified in mice with SJL EAE in our experiments occurred in subarachnoid cisterns, and as a result prominently affected areas of deep gray matter including thalamus and hypothalamus as opposed to cortical lesions also seen in MS. Notably, neuronal loss in the thalamus and other deep gray nuclei does occur in MS with a similar 'surface-in' gradient of neuronal injury, thought to also be related to toxic CSF-derived factors (*Azevedo et al., 2018*; *Magliozzi et al., 2022*). Demyelination, microglia and astrocyte activation, and neuronal loss are evident in the parenchyma adjacent to LMI in SJL EAE (*Bhargava et al., 2021*; *Gupta et al., 2023*). We also noted some variability in the extent and location of LMI, which is unavoidable in EAE. While other animal models of LMI, such as directly injecting inflammatory cytokines into the meninges/cortex, produce more predictable regions of LMI, they involve traumatic injury to the brain and typically lack follicle-like structures (*Silva et al., 2021*). Notably, novel rodent models of cytokine-mediated LMI have recently been developed, including virus-mediated meningeal overexpression of TNF and IFNγ or sub-arachnoid injection of recombinant lymphotoxin-α, were recently developed, and exhibit lymphoid-like structures (*James Bates et al., 2022*; *James et al., 2020*). Spatially resolved analyses of these models could provide additional insights into sub-pial pathology during neuroinflammation.

This work is the first to characterize a mouse model of LMI and gray matter injury using spatial transcriptomics, and in addition to the analysis presented here contributes a publicly available dataset for future research. We highlight the importance of antigen processing and presentation and complement signaling, which are prominently upregulated in sub-pial gray matter, in our model. Future studies should focus on spatial transcriptomics in post-mortem or biopsied human tissue. While access to appropriate samples remains a significant barrier, recent advances in RNA extraction from formalin fixed paraffin embedded tissues (*Newton et al., 2020*; *Gracia Villacampa et al., 2021*) will allow for the use of large banks of historically collected and preserved samples and could dramatically improve availability.

## Materials and methods
### Animals
SJL/J mice were purchased from Jackson Laboratories for all experiments. All mice were maintained in a federally approved animal facility at Johns Hopkins University in accordance with the Institutional Animal Care and Use Committee (protocol # M021 M376). Female mice aged 7–8 weeks were used in all experiments and were housed in the animal facility for at least 1 week prior to the start of experiments.

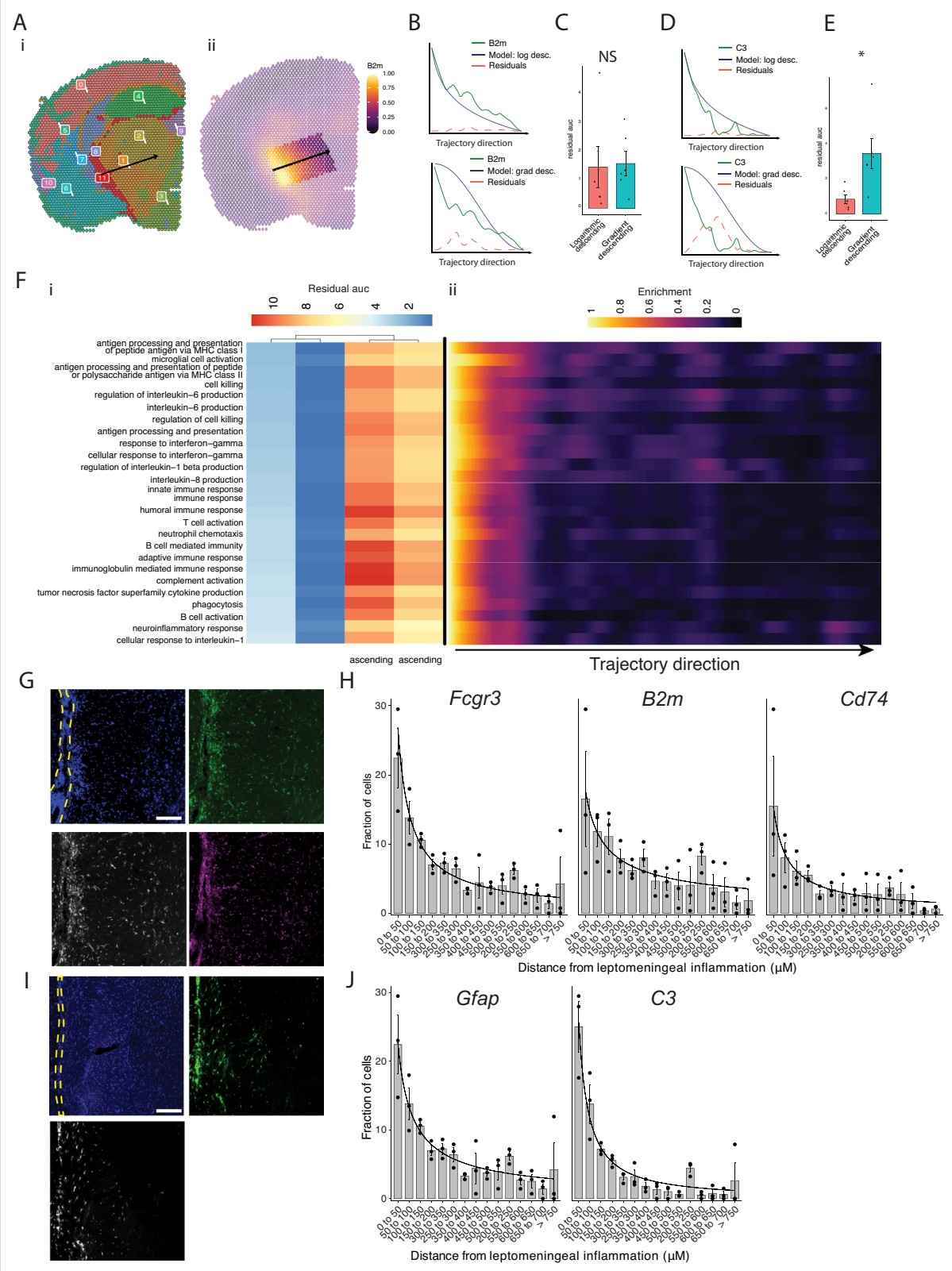

**Figure 5.** Trajectory analysis reveals gradients of gene expression originating from meningeal lymphoid follicles. (**A**) Trajectories were drawn based on spatial cluster plot (**i**) from C11 to C2 (**ii**). (**B**) Representative plot of *B2m* relative expression along the trajectory length. Green line: *B2m* expression; black line: ideal model fit, 'logarithmic descending' (top) or 'gradient descending' (bottom); red line: residual area under the curve (AUC) representing the difference between *B2m* expression and the ideal model. (**C**) Bar plot showing residual AUC of *B2m* relative expression along the

*Figure 5 continued on next page*

*Figure 5 continued*

trajectory direction compared to 'logarithmic descending' or 'gradient descending' (Student's two-tailed t-test). (**D**) Representative plot of *C3* relative expression along the trajectory length. Green line: *C3* expression; black line: ideal model fit, 'logarithmic descending' (top) or 'gradient descending' (bottom); red line: residual area under the curve (AUC) representing the difference between *C3* expression and the ideal model. (**C**) Bar plot showing residual AUC of *C3* relative expression along the trajectory direction compared to 'logarithmic descending' or 'gradient descending' (Student's t-test). (**F**) Gene sets that were previously identified as significantly enriched in C11 were selected for trajectory analysis. Residual AUCs were calculated for 'logarithmic descending', 'gradient descending', 'logarithmic ascending', and 'gradient ascending' ideal fits and displayed on (i) a heatmap sorted by 'gradient descending'. (ii) Representative feature plot demonstrating deeper penetration of upper gene sets (related to antigen presentation and processing, microglial activation, IL-6 production, interferon gamma) response relative to other gene sets (B cell activation, T cell activation, TNF production, complement, humoral immune response). (**G, I**) Representative images of RNAscope labeling for (**G**) *Fcgr3, B2m, Cd74,* and (**I**) *Gfap, C3* in Swiss Jim Lambert (SJL) mice 11 weeks after experimental autoimmune encephalomyelitis (EAE) induction. Yellow dashed lines indicate the areas of leptomeningeal inflammation, scale bars represent 100 μM. (**H, J**) Bar plots representing the percent of marker-positive cells present at distances from leptomeningeal inflammation. Lines represent best fit curves from exponential regression. N=3 animals per group; bars represent mean, error bars represent standard error.

The online version of this article includes the following figure supplement(s) for figure 5:

**Figure supplement 1.** Representative images of RNAscope labeling in naïve and experimental autoimmune encephalomyelitis (EAE) tissues.

## Induction of SJL EAE

Female SJL/J mice were immunized subcutaneously at two sites over the lateral abdomen with 100 μg PLP$_{139-151}$ peptide with complete Freund's adjuvant containing 4 μg/ml *Mycobacterium tuberculosis* H37RA (Difco Laboratories). Mice were weighed and scored serially to document disease course. Scoring was performed using the following scale: 0, normal; 1, limp tail; 2, hind limb weakness; 3, hind limb paralysis; 4, hind limb and forelimb weakness; and 5, death. CFA only controls received CFA without PLP$_{139-151}$ peptide.

## Magnetic resonance imaging

At weeks 6, 8, and 10 post-immunization, a horizontal 11.7 T scanner (Bruker BioSpin) with a triple-axis gradient system (maximum gradient strength = 740 mT/m), 72 mm volume transmit coil, and four-channel receive-only phased array coil was used to image the mouse forebrain. During imaging, mice were anesthetized with isoflurane together with mixed air and oxygen (3:1 ratio) and respiration was monitored via a pressure sensor and maintained at 60 breaths/min. Before imaging, 0.1 ml diluted Magnevist (gadopentetate dimeglumine, Bayer HealthCare LLC, 1:10 with PBS) was injected. Scans were then analyzed to identify areas of meningeal contrast enhancement by two independent examiners (PB, SK). We counted the number of areas of meningeal contrast enhancement on each individual MRI slice and used the cumulative number to represent the amount of meningeal contrast enhancement. All quantifications were performed by at least two independent examiners and their scores were averaged.

## Immunofluorescence preparation and analysis

At 11 weeks' post-injection, CFA only and age-matched naïve animals were perfused with 1× PBS followed by 4% PFA. Brains were dissected, post-fixed for 48 hr, and then cryopreserved in 30% sucrose for 48 hr. Tissue was frozen and then sectioned (12 μm thick) directly onto slides using a cryostat. For immunofluorescent staining, slices were first blocked in blocking buffer (5% normal goat serum in 0.4% Triton X-100 in PBS) for 1 hr at room temperature, followed by overnight incubation in primary antibody. The next day, samples were washed three times in PBS and then incubated with secondary antibody for 1 hr at room temperature. Primary antibodies used in this study were Iba-1 at 1:300 dilution (Cat. No. 019-19741, Wako Chemicals), CD68 at 1:300 dilution (Cat. No. 14-0681-82, Invitrogen), and GFAP at 1:400 dilution (Cat No. GA52461-2, Agilent Dako). Secondary antibodies were Alexa fluorophores (Cat No. A-21244 and A-11006, Invitrogen), all at 1:1000 dilution.

Images were acquired on a Zeiss Axio Observer Z1 epifluorescence microscope and analyzed using FIJI (NIH). Images were centered at the interpeduncular and quadrigeminal cisterns, where LMI typically develops in SJL-EAE, and included the lateral thalamus. Images were acquired and analyzed by an experimenter blinded to the group of each sample and data presented in the form of MFI.

## Tissue preparation and spatial gene expression assay

At 11 weeks' post-immunization, animals were euthanized in $CO_2$ chamber and perfused with cold PBS. Brains were dissected and one hemisphere drop fixed in isopentane cooled on dry ice. Fresh frozen brain samples were then cut coronally at a thickness of 10 µm and placed on the capture area of Visium Gene Expression slides (v.1; 10x Genomics). Each slide contained four 6.5 mm × 6.5 mm capture areas. Sample preparation was carried out according to the manufacturer's instructions. After fixation with methanol at –20°C, H&E staining was performed for morphological analysis and spatial alignment of the sequencing data. After the enzymatic permeabilization, mRNA was captured by probes and cDNA generated. Barcoded cDNA was isolated using SPRIselect-cleanup (Beckman Coulter) and amplified. Amplified cDNA was fragmented and subjected to end-repair, poly-A-tailing, adaptor ligation, and 10×-specific sample indexing as per the manufacturer's instructions. Following assessment of RNA quality, sequencing was performed using a NovaSeq S2 100. Brains from four naïve and four EAE mice were used to prepare five and six individual slices per group, respectively, with once naïve mouse contributing two slices and two EAE mice contributing two slices.

## Spatial transcriptomics data processing

Each sample went through identical quality control processing steps. SpaceRanger software (v.1.3.1) was used to pre-process the sequencing data, aligning to the mm10-2020A reference transcriptome. Feature barcoded expression matrices were used as input for downstream spatial transcriptomics analysis using Seurat (v.4.3.0) and SPATA2 (v.0.1.0) (*Kueckelhaus et al., 2020*). Data was loaded and analyzed using the Seurat, and all spots that were determined to not be over tissue were discarded using the filter.matrix option in Seurat's Load10XSpatial function. Spots with less than 250 measured genes and less than 500 unique molecular identifiers were filtered out. Data normalization and stabilization of sequence depth variance was performed on each sample using SCTransform with default parameters (*Hafemeister and Satija, 2019*). Sample data was then annotated and combined into a merged object for downstream quality control and analysis. Dimensionality reduction was performed using principal component analysis, followed by computation of shared nearest neighbors of the first 10 principal components and cluster identification (resolution 0.3). To visualize all spots in a two-dimensional plot, a UMAP was created with Seurat's RunUMAP function using the first 10 principal components. Cluster and subcluster enriched genes were identified with Wilcoxon tests as implemented in the FindMarkers function within Seurat. Differential gene expression between groups was attained using DESeq2 (v.1.38.2) on samples pseudobulked by biological replicate. Gene set enrichment analysis and visualization was performed using the GO database (*Carbon et al., 2021*; *Ashburner et al., 2000*) and the clusterProfiler package (v.4.6.0) (*Wu et al., 2021*). Estimated signaling pathway activities were calculated for each spot with the top 500 genes of each pathway on SCTransformed data using the PROGENy package (v.1.20.0) (*Schubert et al., 2018*). For subclustering analysis, select clusters were subsetted before undergoing dimensionality reduction, neighbor calculation, cluster identification, marker identification, and gene set enrichment as described above. Trajectory gene and gene set modeling analysis was performed with the SPATA2 package. Spatial trajectories were drawn in EAE slices from the center of cluster 11 to the centro-medial nucleus of thalamus. Gene and gene sets were analyzed along these trajectories using the assessTrajectoryTrends function within SPATA2.

## RNAscope

Brain tissue was collected from N=3 naïve or EAE (11 weeks' post-immunization) SJL mice after transcardial perfusion with 4% PFA. Brains were post-fixed for 48 hr and then dehydrated via sucrose gradient over 48 hr before freezing in OCT. Fourteen-µm-thick tissue sections were collected directly onto Superfrost+ slides (Fisherbrand, Cat. No. 22-037-246). The RNAscope assay was then carried out using the Multiplex Fluorescent Reagent Kit v.2 (Advanced Cell Diagnostics, Cat. No. 323100) per the manufacturer's instructions. Briefly, tissue sections underwent dehydration in an ethanol gradient, peroxide blocking, and antigen retrieval steps. Antigen retrieval was performed using a steamer at 100°C and the epitope retrieval solution provided by the manufacturer. The mRNA in the tissue was hybridized to RNAscope probes (Advanced Cell Diagnostics) against: *C3* (Cat. No. 417841-C3), *Fcgr3* (Cat. No. 587241), *B2m* (Cat. No. 415191-C2), *Cd74* (Cat. No. 437501-C3), and *Gfap* (Cat. No. 313211) for 2 hr at 40°C and stored overnight in 5× saline sodium citrate. Next, amplification steps

were performed according to the manufacturer's instructions. Fluorescent labeling was performed with TSA Vivid Fluorophore 520, TSA Vivid Fluorophore 570, and TSA Vivid Fluorophore 650 (all at 1:2000 dilution). Slides were washed and mounted with ProLong Gold Antifade Mountant (Thermo Fisher, Cat. No. P36830). Slides were imaged via confocal microscopy (Ziess LSM900).

Positive cells were manually labeled in ImageJ (NIH). The x-y coordinate of each cell was used to determine the shortest distance to areas of LMI in R. Distances were batched at intervals of 50 µM. Non-linear regression of the batched data was computed using the stats::nls function and the formula *percentage ~a * distance ^ b*.

## RNAseq statistics and data visualization

Plots were generated and statistics calculated using R (v.4.2.2) and Rstudio (v.2022.07.2). Bar plots, box plots, MA plots, Venn diagram, and dot plots were produced with ggplot2 (v.3.4.0); heatmaps were produced with pHeatmap (v.1.0.12) and enrichplot (v.1.18.4) (*Guangchuang, 2023*); spatial feature plots and dimensionality reduction plots were produced with Seurat; tree plots were produced with enrichplot, and trajectory heatmaps were produced with SPATA2.

All reported p-values adjusted for multiple comparisons were corrected using the Benjamini-Hochberg method (*Benjamini and Hochberg, 1995*) unless otherwise specified. The number of samples in the EAE and naïve groups was chosen based on expected levels of variability in prior experiments.

## Acknowledgements

We thank members of the Calabresi lab for their valuable comments during multiple discussions of this work. We thank the Johns Hopkins Medicine Single Cell and Transcriptomics Core for their assistance with planning and data acquisition. We acknowledge the contribution of animals used in this research study. This work was supported in part by an investigator-initiated grant from EMD-Serono to PB, a Harry Weaver Neuroscience Scholar award to PB, and a fellowship grant from the National Multiple Sclerosis Society and the American Brain Foundation (FAN-2106–37832) to SPG.

## Additional information

### Funding

| Funder | Grant reference number | Author |
|---|---|---|
| National Multiple Sclerosis Society | FAN-2106-37832 | Sachin P Gadani |
| National Multiple Sclerosis Society | JF-2007-36755 | Pavan Bhargava |
| EMD Serono | | Pavan Bhargava |

The funders had no role in study design, data collection and interpretation, or the decision to submit the work for publication.

### Author contributions

Sachin P Gadani, Data curation, Formal analysis, Investigation, Visualization, Writing – original draft, Writing – review and editing; Saumitra Singh, Data curation, Formal analysis, Validation, Investigation, Methodology, Writing – original draft, Writing – review and editing; Sophia Kim, Investigation, Writing – review and editing; Jingwen Hu, Validation, Investigation, Writing – review and editing; Matthew D Smith, Data curation, Formal analysis, Validation, Visualization, Writing – review and editing; Peter A Calabresi, Supervision, Writing – review and editing; Pavan Bhargava, Conceptualization, Data curation, Supervision, Funding acquisition, Investigation, Writing – original draft, Project administration, Writing – review and editing

### Author ORCIDs

Sachin P Gadani http://orcid.org/0000-0002-9555-4553
Peter A Calabresi https://orcid.org/0000-0002-7776-6472
Pavan Bhargava https://orcid.org/0000-0002-7947-9418

## Ethics

All animal studies followed national and institutional guidelines for humane animal treatment in compliance with the Johns Hopkins ACUC (protocol # M021 M376).

Reviewer 2 (Public Review): https://doi.org/10.7554/eLife.88414.4.sa1
Reviewer 1 (Public Review): https://doi.org/10.7554/eLife.88414.4.sa2
Author response https://doi.org/10.7554/eLife.88414.4.sa3

---

# Additional files

## Supplementary files

• Supplementary file 1. Results of differentially expressed gene analysis comparing experimental autoimmune encephalomyelitis (EAE) and naive brain slices.

• Supplementary file 2. Results of differentially expressed gene analysis comparing cluster 11 with other clusters.

• Supplementary file 3. Results of gene set enrichment analysis based on cluster 11 differentially expressed genes.

• Supplementary file 4. Results of gene set enrichment analysis based on cluster 1_3 differentially expressed genes.

• Supplementary file 5. Results of gene set enrichment analysis based on cluster 1_4 differentially expressed genes.

• Supplementary file 6. Results of gene set enrichment analysis based on cluster 2_6 differentially expressed genes.

• Supplementary file 7. Exponential regression analysis of RNAscope data.

## Data availability

Spatial transcriptomics sequencing data has been deposited in the Gene Expression Omnibus (GEO) database under accession code GSE236963.

The following dataset was generated:

| Author(s) | Year | Dataset title | Dataset URL | Database and Identifier |
|---|---|---|---|---|
| Gadani SP, Singh S, Smith M, Kim S, Hu J, Calabresi PA, Bhargava P | 2024 | Spatial Transcriptomics of Meningeal Inflammation Reveals Variable Penetrance of Inflammatory Gene Signatures into Adjacent Brain Parenchyma | https://www.ncbi.nlm.nih.gov/geo/query/acc.cgi?acc=GSE236963 | NCBI Gene Expression Omnibus, GSE236963 |

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
