## [Editor Report · eLife Assessment]

Brain inflammation is a hallmark of multiple sclerosis. Using novel spatial transcriptomics methods, the authors provide **solid** evidence for a gradient of immune genes and inflammatory markers from the meninges toward the adjacent brain parenchyma in a mouse model. This **important** study advances our understanding of the mechanisms of brain damage in this autoimmune disease.

---

## [Referee Report · Reviewer 2 (Public Review)]

Accumulating data suggests that the presence of immune cell infiltrates in the meninges of the multiple sclerosis brain contributes to the tissue damage in the underlying cortical grey matter by the release of inflammatory and cytotoxic factors that diffuse into the brain parenchyma. However, little is known about the identity and direct and indirect effects of these mediators at a molecular level. This study addresses the vital link between an adaptive immune response in the CSF space and the molecular mechanisms of tissue damage that drive clinical progression. In this short report the authors use a spatial transcriptomics approach using Visium Gene Expression technology from 10x Genomics, to identify gene expression signatures in the meninges and the underlying brain parenchyma, and their interrelationship, in the PLP-induced EAE model of MS in the SJL mouse. MRI imaging using a high field strength (11.7T) scanner was used to identify areas of meningeal infiltration for further study. They report, as might be expected, the upregulation of genes associated with the complement cascade, immune cell infiltration, antigen presentation, and astrocyte activation. Pathway analysis revealed the presence of TNF, JAK-STAT and NFkB signaling, amongst others, close to sites of meningeal inflammation in the EAE animals, although the spatial resolution is insufficient to indicate whether this is in the meninges, grey matter, or both.

UMAP clustering illuminated a major distinct cluster of upregulated genes in the meninges and smaller clusters associated with the grey matter parenchyma underlying the infiltrates. The meningeal cluster contained genes associated with immune cell functions and interactions, cytokine production, and action. The parenchymal clusters included genes and pathways related to glial activation, but also adaptive/B-cell mediated immunity and antigen presentation. This again suggests a technical inability to resolve fully between the compartments as immune cells do not penetrate the pial surface in this model or in MS. Finally, a trajectory analysis based on distance from the meningeal gene cluster successfully demonstrated descending and ascending gradients of gene expression, in particular a decline in pathway enrichment for immune processes with distance from the meninges.

---

## [Referee Report · Reviewer 1 (Public Review)]

Multiple sclerosis (MS) is a debilitating autoimmune disease that causes loss of myelin in neurons of the central nervous system. MS is characterized by the presence of inflammatory immune cells in several brain regions as well as the brain barriers (meninges). This study aims to understand the local immune hallmarks in regions of the brain parenchyma that are adjacent to the leptomeninges in a mouse model of MS. The leptomeninges are known to be a foci of inflammation in MS and perhaps "bleed" inflammatory cells and molecules to adjacent brain parenchyma regions. To do so, they use novel technology called spatial transcriptomics so that the spatial relationships between the two regions remain intact. The study identifies canonical inflammatory genes and gene sets such as complement and B cells enriched in the parenchyma in close proximity to the leptomeninges in the mouse model of MS but not control. The manuscript is very well written and easy to follow. The results will become a useful resource to others working in the field and can be followed by time series experiments where the same technology can be applied to the different stages of the disease.

---

## [Author Response]

The following is the authors’ response to the previous reviews

**Reviewer 1 (Public Review):**
Multiple sclerosis (MS) is a debilitating autoimmune disease that causes loss of myelin in neurons of the central nervous system. MS is characterized by the presence of inﬂammatory immune cells in several brain regions as well as the brain barriers (meninges). This study aims to understand the local immune hallmarks in regions of the brain parenchyma that are adjacent to the leptomeninges in a mouse model of MS. The leptomeninges are known to be a foci of inﬂammation in MS and perhaps "bleed" inﬂammatory cells and molecules to adjacent brain parenchyma regions. To do so, they use novel technology called spatial transcriptomics so that the spatial relationships between the two regions remain intact. The study identiﬁes canonical inﬂammatory genes and gene sets such as complement and B cells enriched in the parenchyma in close proximity to the leptomeninges in the mouse model of MS but not control. The manuscript is very well written and easy to follow. The results will become a useful resource to others working in the ﬁeld and can be followed by time series experiments where the same technology can be applied to the diAerent stages of the disease.Comments on revised version:I agree that the authors successfully addressed most of my comments/critiques. However, the fact that the control mice were not injected with CFA and pertussis toxin is somewhat concerning, because it will be hard to interpret the cause of the transcriptomic readouts described in this study. Some of the described eAects might be due to CFA or pertussis (which was used in the EAE but not the "naive" group), and not necessarily to the relapsing-remitting EAE immune features recapitulated in this mouse model. Moreover, this caveat associated with the "naive" control group is not being clearly stated throughout the manuscript and might go unnoticed to readers.The authors should clearly state, in the methods section (in the section "Induction of SJL EAE"), that the naive control group was not injected with CFA or pertussis toxin.Additionally, this potential confounder, of not using a control group injected with the same CFA and pertussis toxin regimen of the EAE group, should be mentioned in paragraph two of the discussion alongside the other limitations of the study already highlighted by the authors (or in another section of the discussion).

We thank the reviewer for highlighting this point. Our choice of healthy/naïve, rather than CFA only, controls was intentional, given our desire to sensitively measure genes changing during neuroinflammation. Ultimately, however, we believe the choice of control group had little effect on our conclusions. We would like to note that SJL-EAE does not require pertussis toxin, so the only difference between naïve and CFA only groups is a single injection of CFA 11 weeks prior to experiment endpoint. We have performed additional IHC imaging of naïve and CFA only groups, finding no difference in glial reactivity by MFI measurement of GFAP, IBA1, or CD68 (updated Supplementary Figure 1C–E).

We have also added sections to the Results and Discussion section to clearly address this point. In the Results: “Since naïve animals were used as controls, we confirmed that CFA alone does not produce lasting glial reactivity or LMI formation. Groups of animals were given CFA only or left naïve. Neither group developed neurologic signs, and after 11 weeks the brains were processed for IHC analysis. There was no evidence of LMI development, and no difference in glial reactivity as measured by GFAP, IBA1, or CD68 intensity (Supplemental Figure 1C–E).” In the Discussion: “Another important consideration in these experiments is our choice of naïve, rather than CFA only, controls. While often used as the control in EAE studies focused on mechanisms of autoimmunity, CFA only can independently induce systemic inflammation. Since this study seeks to describe transcriptomic changes in neuroinflammation more broadly, we chose to use a healthy comparison group to maximize our ability to find genes enriched in neuroinflammation. Ultimately, however, the choice of naïve or CFA only controls is unlikely to have affected our conclusions. SJL-EAE, unlike the more common C57Bl6-EAE, does not require pertussis toxin during the induction. The only difference between naïve and CFA only controls is the subcutaneous CFA delivered at time of immunization (11 weeks prior to experiment endpoint). Indeed, when we compared CFA only and healthy animals at 11 weeks there was no difference in glial reactivity by GFAP, IBA1, or CD68 MFI. There was also no evidence of neurologic symptoms or LMI development in CFA only controls.”

**Reviewer 2 (Public Review):**
Accumulating data suggests that the presence of immune cell inﬁltrates in the meninges of the multiple sclerosis brain contributes to the tissue damage in the underlying cortical grey matter by the release of inﬂammatory and cytotoxic factors that diAuse into the brain parenchyma. However, little is known about the identity and direct and indirect eAects of these mediators at a molecular level. This study addresses the vital link between an adaptive immune response in the CSF space and the molecular mechanisms of tissue damage that drive clinical progression. In this short report the authors use a spatial transcriptomics approach using Visium Gene Expression technology from 10x Genomics, to identify gene expression signatures in the meninges and the underlying brain parenchyma, and their interrelationship, in the PLP-induced EAE model of MS in the SJL mouse. MRI imaging using a high ﬁeld strength (11.7T) scanner was used to identify areas of meningeal inﬁltration for further study. They report, as might be expected, the upregulation of genes associated with the complement cascade, immune cell inﬁltration, antigen presentation, and astrocyte activation. Pathway analysis revealed the presence of TNF, JAK-STAT and NFkB signaling, amongst others, close to sites of meningeal inﬂammation in the EAE animals, although the spatial resolution is insuAicient to indicate whether this is in the meninges, grey matter, or both.UMAP clustering illuminated a major distinct cluster of upregulated genes in the meninges and smaller clusters associated with the grey matter parenchyma underlying the inﬁltrates. The meningeal cluster contained genes associated with immune cell functions and interactions, cytokine production, and action. The parenchymal clusters included genes and pathways related to glial activation, but also adaptive/B-cell mediated immunity and antigen presentation. This again suggests a technical inability to resolve fully between the compartments as immune cells do not penetrate the pial surface in this model or in MS. Finally, a trajectory analysis based on distance from the meningeal gene cluster successfully demonstrated descending and ascending gradients of gene expression, in particular a decline in pathway enrichment for immune processes with distance from the meninges.Comments on revised version:The authors have addressed all of my comments regarding the lack of spatial resolution between the grey matter and the overlying meninges and also concerning the diAiculties in extrapolating from this mouse model to MS itself.I am however very concerned about the lack of the correct control group. Immunization of rodents with complete freunds adjuvant and pertussis alone gives rise to widespread microglial activation, some immune cell inﬁltration and also structural changes to axons, particularly at nodes of Ranvier (https://doi.org/10.1097/NEN.0b013e3181f3a5b1). This will inevitably make it diAicult to interpret the transcriptomics results, depending on whether these changes are reversible or not and the time frame of the reversal. In the C57Bl6 EAE models adjuvant induced microglial activation becomes chronic, whereas the axonal changes do reverse by 10 weeks. Whether this is the same in SJL EAE model is not clear.

We thank the reviewer for bringing up this concern regarding control group, which we discussed above in point 1.1. To speciﬁcally address reviewer 2’s point regarding microglial activation, we performed IHC analysis comparing naïve and CFA only groups of SJL animals. We found no substantial diAerence in astrocyte or microglial activation in these animals after 11 weeks, as measured by GFAP, IBA1, and CD68. This new data appears in updated Supplementary Figure 1C–D.

**Recommendations for the authors:**
Both reviewers agree that the revised version has improved and some of their major concerns were adequately addressed. However, both reviewers also agree that critical experimental controls are missing, including the FCA and pertussis toxin injected mice which likely show some degree of inﬂammation in their brain and are needed to compare your experimental MS group and interpret the transcriptomics data.

We appreciate both reviewers’ important comments on the control group used in this study. In this revised manuscript we have described our rationale for choosing naïve controls, rather than CFA only, and believe they are the most appropriate comparison group. Additionally, we believe that both CFA only and naïve will have similar degrees of baseline neuroinflammation at the 11- week time point. We apologize for not clarifying before, but pertussis toxin is not used in the SJL-EAE, and therefore the “CFA only” control is much milder in SJL-EAE compared to C57Bl6-EAE. Given that many signs of inflammation resolve by 10 weeks in CFA only with pertussis controls (https://academic.oup.com/jnen/article/69/10/1017/2917071; https://www.ncbi.nlm.nih.gov/pmc/articles/PMC10902151/), CFA only without pertussis controls are unlikely to have any substantial remaining neuroinflammation at 11 weeks. To test this, we performed an additional experiment directly comparing naïve and CFA only without pertussis.

These groups showed similar degrees of glial reactivity.

Given the costs of repeating a spatial transcriptomic experiment and inevitable batch effects should we add a group at this point, we have chosen to not as a CFA only control condition to our transcriptomics analysis. However, we believe our added text clarifying the rationale behind control choice and added immunofluorescence data gives readers the appropriate context to accurately interpret our results.